# Teachable Reinforcement Learning via Advice Distillation

**Olivia Watkins**
UC Berkeley
oliviawatkins@berkeley.edu

**Trevor Darrell**
UC Berkeley
trevor@eecs.berkeley.edu

**Pieter Abbeel**
UC Berkeley
pabbeel@cs.berkeley.edu

**Jacob Andreas**
MIT
jda@mit.edu

**Abhishek Gupta**
UC Berkeley
abhigupta@berkeley.edu

## Abstract

Training automated agents to complete complex tasks in interactive environments is challenging: reinforcement learning requires careful hand-engineering of reward functions, imitation learning requires specialized infrastructure and access to a human expert, and learning from intermediate forms of supervision (like binary preferences) is time-consuming and extracts little information from each human intervention. Can we overcome these challenges by building agents that learn from rich, interactive feedback instead? We propose a new supervision paradigm for interactive learning based on "teachable" decision-making systems that learn from structured advice provided by an external teacher. We begin by formalizing a class of human-in-the-loop decision making problems in which multiple forms of teacher-provided advice are available to a learner. We then describe a simple learning algorithm for these problems that first *learns to interpret advice*, then *learns from advice* to complete tasks even in the absence of human supervision. In puzzle-solving, navigation, and locomotion domains, we show that agents that learn from advice can acquire new skills with significantly less human supervision than standard reinforcement learning algorithms and often less than imitation learning.

## 1 Introduction

Reinforcement learning (RL) offers a promising paradigm for building agents that can learn complex behaviors from autonomous interaction and minimal human effort. In practice, however, significant human effort is required to design and compute the reward functions that enable successful RL [49]: the reward functions underlying some of RL's most prominent success stories involve significant domain expertise and elaborate instrumentation of the agent and environment [37, 38, 44, 28, 15]. Even with this complexity, a reward is ultimately no more than a scalar indicator of how good a particular state is relative to others. Rewards provide limited information about *how* to perform tasks, and reward-driven RL agents must perform significant exploration and experimentation within an environment to learn effectively. A number of alternative paradigms for interactively learning policies have emerged as alternatives, such as imitation learning [40, 20, 50], DAgger [43], and preference learning [10, 6]. But these existing methods are either impractically low bandwidth (extracting little information from each human intervention) [25, 30, 10] or require costly data collection [44, 23]. It has proven challenging to develop training methods that are simultaneously expressive and efficient enough to rapidly train agents to acquire novel skills.

Human learners, by contrast, leverage numerous, rich forms of supervision: joint attention [34], physical corrections [5] and natural language instruction [9]. For human teachers, this kind of

35th Conference on Neural Information Processing Systems (NeurIPS 2021), virtual.

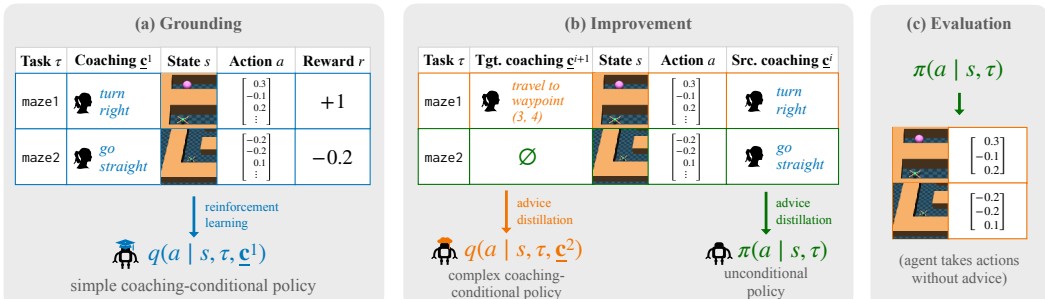

Figure 1: Three phases of teachable reinforcement learning. During the **grounding** phase (a), we train an advice-conditional policy through RL $q(a|s, \tau, c^1)$ that can interpret a simple form of advice $c^1$. During the **improvement** phase (b), an external coach provides real-time coaching, which the agent uses to learn more complex advice forms and ultimately an advice-independent policy $\pi(a|s, \tau)$. During the **evaluation** phase, the advice-independent policy $\pi(a|s, \tau)$ is executed to accomplish a task without additional human feedback.

coaching is often no more costly to provide than scalar measures of success, but significantly more informative for learners. In this way, human learners use high-bandwidth, low-effort communication as a means to flexibly acquire new concepts or skills [46, 33]. Importantly, the interpretation of some of these feedback signals (like language), is itself learned, but can be *bootstrapped* from other forms of communication: for example, the function of gesture and attention can be learned from intrinsic rewards [39]; these in turn play a key role in language learning [31].

This paper proposes a framework for training automated agents using similarly rich interactive supervision. For instance, given an agent learning a policy to navigate and manipulate objects in a simulated multi-room object manipulation problem (e.g., Fig 3 left), we train agents using not just reward signals but advice about what actions to take ("move left"), what waypoints to move towards ("move towards $(1, 2)$"), and what sub-goals to accomplish ("pick up the yellow ball"), offering human supervisors a toolkit of rich feedback forms that direct and modify agent behavior. To do so, we introduce a new formulation of interactive learning, the Coaching-Augmented Markov Decision Process (CAMDP), which formalizes the problem of learning from a privileged supervisory signal provided via an observation channel. We then describe an algorithmic framework for learning in CAMDPs via alternating advice *grounding* and *distillation* phases. During the grounding phase, agents learn associations between teacher-provided advice and high-value actions in the environment; during distillation, agents collect trajectories with grounded models and interactive advice, then transfer information from these trajectories to fully autonomous policies that operate without advice. This formulation allows supervisors to guide agent behavior interactively, while enabling agents to internalize this guidance to continue performing tasks autonomously once the supervisor is no longer present. Moreover, this procedure can be extended to enable *bootstrapping* of grounded models that use increasingly sparse and abstract advice types, leveraging some types of feedback to ground others. Experiments show that models trained via coaching can learn new tasks more efficiently and with 20x less human supervision than naïve methods for RL across puzzle-solving [8], navigation [14], and locomotion domains [8].

In summary, this paper describes: (1) a general framework (CAMDPs) for human-in-the-loop RL with rich interactive advice; (2) an algorithm for learning in CAMDPs with a single form of advice; (3) an extension of this algorithm that enables bootstrapped learning of multiple advice types; and finally (4) a set of empirical evaluations on discrete and continuous control problems in the BabyAI [8] and D4RL [14] environments. It thus offers a groundwork for moving beyond reward signals in interactive learning, and instead training agents with the full range of human communicative modalities.

## 2   Coaching Augmented Markov Decision Processes

To develop our procedure for learning from rich feedback, we begin by formalizing the environments and tasks for which feedback is provided. This formalization builds on the framework of multi-task RL and Markov decision processes (MDP), augmenting them with advice provided by a coach in the loop through an arbitrary prescriptive channel of communication. Conider the grid-world environment depicted in Fig 3 left [8]. **Tasks** in this environment specify particular specific desired goal states; e.g. "place the yellow ball in the green box and the blue key in the green box" or "open all doors in

the blue room." In multi-task RL, a learner's objective is produce a policy $\pi(a_t|s_t, \tau)$ that maximizes reward in expectation over tasks $\tau$. More formally, a **multi-task MDP** is defined by a 7-tuple $\mathcal{M} \equiv (\mathcal{S}, \mathcal{A}, \mathcal{T}, \mathcal{R}, \rho(s_0), \gamma, p(\tau))$, where $\mathcal{S}$ denotes the state space, $\mathcal{A}$ denotes the action space, $p : \mathcal{S} \times \mathcal{A} \times \mathcal{S} \mapsto \mathbb{R}_{\geq 0}$ denotes the transition dynamics, $r : \mathcal{S} \times \mathcal{A} \times \tau \mapsto \mathbb{R}$ denotes the reward function, $\rho : \mathcal{S} \mapsto \mathbb{R}_{\geq 0}$ denotes the initial state distribution, $\gamma \in [0, 1]$ denotes the discount factor and $p(\tau)$ denotes the distribution over tasks. The objective in a multi-task MDP is to learn a policy $\pi_\theta$ that maximizes the expected sum of discounted returns in expectation over tasks: $\max_\theta J_E(\pi_\theta, p(\tau)) = \mathbb{E}_{\substack{a_t \sim \pi_\theta(\cdot|s_t, \tau) \\ \tau \sim p(\tau)}} [\sum_{t=0}^{\infty} \gamma^t r(s_t, a_t, \tau)]$.

Why might additional supervision beyond the reward signal be useful for solving this optimization problem? Suppose the agent in Fig 3 is in the (low-value) state shown in the figure, but could reach a high-value state by going "right and up" towards the blue key. This fact is difficult to communicate through a scalar reward, which cannot convey information about alternative actions. A side channel for providing this type of rich information at training-time would be greatly beneficial.

We model this as follows: a **coaching-augmented MDP (CAMDP)** consists of an ordinary multi-task MDP augmented with a set of **coaching functions** $\mathcal{C} = \{\mathcal{C}^1, \mathcal{C}^2, \cdots, \mathcal{C}^i\}$, where each $C^j$ provides a different form of feedback to the agent. Like a reward function, each coaching function models a form of supervision provided externally to the agent (by a **coach**); these functions may produce informative outputs densely (at each timestep) or only infrequently. Unlike rewards, which give agents feedback on the desirability of states and actions they have already experienced, this coaching provides information about what the agent should do next. [1] As shown in Figure 3, advice can take many forms, for instance action advice ($c^0$), waypoints ($c^1$), language sub-goals ($c^2$), or any other local information relevant to task completion.[2] Coaching in a CAMDP is useful if it provides an agent local guidance on how to proceed toward a goal that is inferrable from the agent's current observation, when the mapping from observations and goals to actions has not yet been learned.

As in standard reinforcement learning in an multi-task MDP, the goal in a CAMDP is to learn a policy $\pi_\theta(\cdot \mid s_t, \tau)$ that chooses an action based on Markovian state $s_t$ and high level task information $\tau$ *without* interacting with $c^j$. However, we allow *learning algorithms* to use the coaching signal $c^j$ to learn this policy more efficiently at training time (although this is unavailable during deployment). For instance, the agent in Fig 3 can leverage hints "go left" or "move towards the blue key" to guide its exploration process but it eventually must learn how to perform the task *without* any coaching required. Section 3 describes an algorithm for acquiring this independent, multi-task policy $\pi_\theta(\cdot \mid s_t, \tau)$ from coaching feedback, and Section 4 presents an empirical evaluation of this algorithm.

## 3 Leveraging Advice via Distillation

### 3.1 Preliminaries

The challenge of learning in a CAMDP is twofold: first, agents must learn to ground coaching signals in concrete behavior; second, agents must learn from these coaching signals to independently solve the task of interest in the absence of any human supervision. To accomplish this, we divide agent training into three phases: (1) a *grounding* phase, (2) an *improvement* phase and (3) an *evaluation* phase.

In the grounding phase, agents learn *how* to interpret coaching. The result of the grounding phase is a surrogate policy $q(a_t \mid s_t, \tau, \mathbf{c})$ that can effectively condition on coaching when it is provided in the training loop. As we will discuss in Section 3.2, this phase can also make use of a *bootstrapping* process in which more complex forms of feedback are learned using signals from simpler ones.

During the improvement phase, agents use the ability to interpret advice to learn new skills. Specifically, the learner is presented with a novel task $\tau_{\text{test}}$ that was not provided during the grounding phase, and must learn to perform this task using only a small amount of interaction in which advice $c$ is provided by a human supervisor who is present in the loop. This advice, combined with the

---

[1]While the design of optimal coaching strategies and explicit modeling of coaches are important research topics [16], this paper assumes that the coach is fixed and not explicitly modeled. Our empirical evaluation use both scripted coaches and human-in-the-loop feedback.

[2]When only a single form of advice is available to the agent, we omit the superscript for clarity.

learned surrogate policy $q(a_t|s_t, \tau, \underline{\mathbf{c}})$, can be used to efficiently acquire an advice-*independent* policy $\pi(a_t|s_t, \tau)$, which can perform tasks without requiring any coaching.

Finally, in the evaluation phase, agent performance is evaluated on the task $\tau_{\text{test}}$ by executing the advice-independent, multi-task policy $\pi(a_t|s_t, \tau_{\text{test}})$in the environment.

## 3.2 Grounding Phase: Learning to Interpret Advice

The goal of the grounding phase is to learn a mapping from advice to contextually appropriate actions, so that advice can be used for quickly learning new tasks. In this phase, we run RL on a distribution of training tasks $p(\tau)$. As the purpose of these training environments is purely to ground coaching, sometimes called "advice", the tasks may be much simpler than test-time tasks. During this phase, the agent uses access to a reward function $r(s, a, c)$, as well as the advice $c(s, a)$ to learn a surrogate policy $q_\phi(a|s, \tau, c)$. The reward function $r(s, a, c)$ is provided by the coach during the grounding phase only and rewards the agent for correctly following the provided coaching, not just for accomplishing the task. Since coaching instructions (e.g. cardinal directions) are much easier to follow than completing a full task, grounding can be learned quickly. The process of grounding is no different than standard multi-task RL, incorporating advice $c(s, a)$ as another component of the observation space. This formulation makes minimal assumptions about the form of the coaching $c$.

During this grounding process, the agent's optimization objective is:

$$\max_\phi J(\theta) = \mathbb{E}_{\substack{\tau \sim p(\tau) \\ a_t \sim q_\phi(a_t|s_t, \tau, c)}} \left[ \sum_t r(s_t, a_t, c) \right], \tag{1}$$

**Bootstrapping Multi-Level Advice**   The previous section described how to train an agent to interpret a single form of advice $c$. In practice, a coach might find it useful to use multiple forms of advice—for instance high-level language sub-goals for easy stages of the task and low-level action advice for more challenging parts of the task. While high-level advice can be very informative for guiding the learning of new tasks in the improvement phase, it can often be quite difficult to ground quickly pure RL. Instead of relying on RL, we can bootstrap the process of grounding one form of advice $c^h$ from a policy $q(a|s, \tau, c^l)$ that can interpret a different form of advice $c^l$. In particular, we can use a surrogate policy which already understands (using the grounding scheme described above) low-level advice $q(a|s, \tau, c^l)$ to bootstrap training of a surrogate policy which understands higher-level advice $q(a|s, \tau, c^h)$. We call this process "bootstrap distillation".

Intuitively, we use a supervisor in the loop to guide an advice-conditional policy that can interpret a low-level form of advice $q_{\phi_1}(a|s, \tau, c^l)$ to perform a training task, obtaining trajectories $\mathscr{D} = \{(s_0, a_0, c_0^l, c_0^h), (s_1, a_1, c_1^l, c_1^h) \cdots, (s_H, a_H, c_H^l, c_H^h)\}_{j=1}^N$, then distilling the demonstrated behavior via supervised learning into a policy $q_{\phi_2}(a|s, \tau, c^h)$ that can interpret higher-level advice to perform this

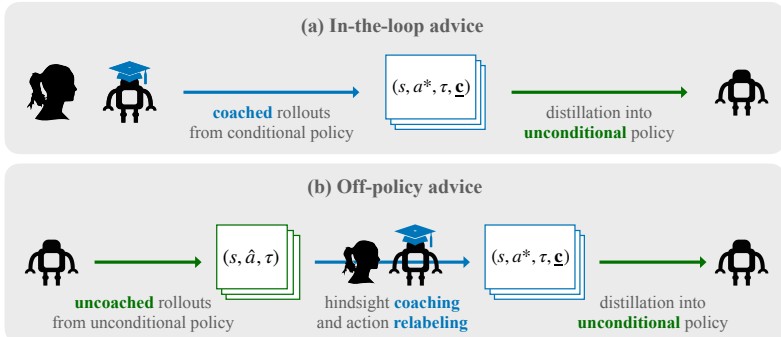

Figure 2: Illustration of the procedure of advice distillation in the on-policy and off-policy settings. During on-policy advice distillation, the advice-conditional surrogate policy $q(a|s, \tau, c)$ is coached to get optimal trajectories. These trajectories are then distilled into an *unconditional* model $\pi(a|s, \tau)$ using supervised learning. During off-policy distillation, trajectories are collected by the unconditional policy and trajectories are relabeled with advice after the fact. After this, we use the advice-conditional policy $q(a|s, \tau, c)$ to relabel trajectories with optimal actions. These trajectories can then be distilled into an unconditional policy.

new task **without** requiring the low level advice any longer. More specifically, we make use of an input remapping solution, as seen in Levine et al. [28], where the policy conditioned on advice $c^l$ is used to generate optimal action labels, which are then remapped to observations with a different form of advice $c^h$ as input. To bootstrap the understanding of an abstract form of advice $c^h$ from a more low level one $c^l$, the agent optimizes the following objective to bootstrap the agent's understanding of one advice type from another:

$$\mathcal{D} = \{(s_0, a_0, c_0^l, c_0^h), (s_1, a_1, c_1^l, c_1^h), \cdots, (s_H, a_H, c_H^l, c_H^h)\}_{j=1}^N$$

$$s_0 \sim p(s_0), a_t \sim q_{\phi_1}(a_t | s_t, \tau, c^l), s_{t+1} \sim p(s_{t+1} | s_t, a_t)$$

$$\max_{\phi_2} \mathbb{E}_{(s_t, a_t, c_t^h, \tau) \sim \mathcal{D}} \left[ \log q_{\phi_2}(a_t | s_t, \tau, c_t^h) \right]$$

With this procedure, we only need to use RL to ground the simplest, fastest-learned advice form, and we can use more efficient bootstrapping to ground the others.

### 3.3 Improvement Phase: Learning New Tasks Efficiently with Advice

At the end of the grounding phase, we have an advice-following agent $q_\phi(a | s, \tau, \underline{\mathbf{c}})$ that can interpret various forms of advice. Ultimately, we want a policy $\pi(a | s, \tau)$ which is able to succeed at performing the new test task $\tau_{\text{test}}$, *without* requiring advice at evaluation time. To achieve this, we make use of a similar idea to the one described above for bootstrap distillation. In the improvement phase, we leverage a supervisor in the loop to guide an advice-conditional surrogate policy $q_\phi(a | s, \tau, c)$ to perform the new task $\tau_{\text{test}}$, obtaining trajectories $\mathcal{D} = \{s_0, a_0, c_0, s_1, a_1, c_1, \cdots, s_H, a_H, c_H\}_{j=1}^N$, then distill this behavior into an advice-independent policy $\pi_\theta(a | s, \tau)$ via behavioral cloning. The result is a policy trained using coaching, but ultimately able to select tasks even when no coaching is provided. In Fig 3 left, this improvement process would involve a coach in the loop providing action advice or language sub-goals to the agent during learning to coach it towards successfully accomplishing a task, and then distilling this knowledge into a policy that can operate without seeing action advice or sub-goals at execution time. More formally, the agent optimizes the following objective:

$$\mathcal{D} = \{s_0, a_0, c_0, s_1, a_1, c_1, \cdots, s_H, a_H, c_H\}_{j=1}^N$$

$$s_0 \sim p(s_0), a_t \sim q_\phi(a_t | s_t, \tau, c_t), s_{t+1} \sim p(s_{t+1} | s_t, a_t)$$

$$\max_\theta \mathbb{E}_{(s_t, a_t, \tau) \sim \mathcal{D}} [\log \pi_\theta(a_t | s_t, \tau)]$$

This improvement process, which we call advice distillation, is depicted Fig 2. This distillation process is preferable over directly providing demonstrations because the advice provided can be more convenient than providing an entire demonstration (for instance, compare the difficulty of producing a demo by navigating an agent through an entire maze to providing a few sparse waypoints). Interestingly, even if the new tasks being solved $\tau_{\text{test}}$ are quite different from the training distribution of tasks $p(\tau)$, since advice $c$ (for instance waypoints) is provided locally and is largely invariant to this distribution shift, the agent's understanding of advice generalizes well.

**Learning with Off-Policy Advice**    One limitation to the improvement phase procedure described above is that advice must be provided in real time. However, a small modification to the algorithm allows us to train with off-policy advice. During the improvement phase, we roll out an initially-untrained advice-independent policy $\pi(a | s, \tau)$. After the fact, the coach provides high-level advice $c^h$ at a multiple points along the trajectory. Next, we use the advice-conditional surrogate policy $q_\phi(a | s, \tau, \underline{\mathbf{c}})$ to relabel this trajectory with near-optimal actions at each timestep. This lets us use behavioral cloning to update the advice-free agent on this trajectory. While this relabeling process must be performed multiple times during training, it allows a human to coach an agent **without providing real-time advice**, which can be more convenient. This process can be thought of as the coach performing DAgger [42] at the level of high-level advice (as was done in in [26]) rather than low-level actions. This procedure can be used for both the grounding and improvement phases. Mathematically, the agent optimizes the following objective:

$$\mathcal{D} = \{s_0, a_0, c_0, s_1, a_1, c_1, \cdots, s_H, a_H, c_H\}_{j=1}^N$$

$$s_0 \sim p(s_0), a_t \sim \pi(a_t | s_t, \tau), s_{t+1} \sim p(s_{t+1} | s_t, a_t)$$

$$\max_\theta \mathbb{E}_{\substack{(s_t, \tau) \sim \mathcal{D} \\ a^* \sim q_\phi(a_t | s_t, \tau, c)}} \left[ \log \pi_\theta(a^* | s_t, \tau) \right]$$

### 3.4 Evaluation Phase: Executing tasks Without a Supervisor

In the evaluation phase, the agent simply needs to be able to perform the test tasks $\tau_{\text{test}}$ without requiring a coach in the loop. We run the advice-independent agent learned in the improvement phase, $\pi(a|s, \tau)$ on the test task $\tau_{\text{test}}$ and record the average success rate.

## 4 Experimental Evaluation

We aim to answer the following questions through our experimental evaluation (1) Can advice be grounded through interaction with the environment via supervisor in the loop RL? (2) Can grounded advice allow agents to learn new tasks more efficiently than standard RL? (3) Can agents bootstrap the grounding of one form of advice from another?

### 4.1 Evaluation Domains

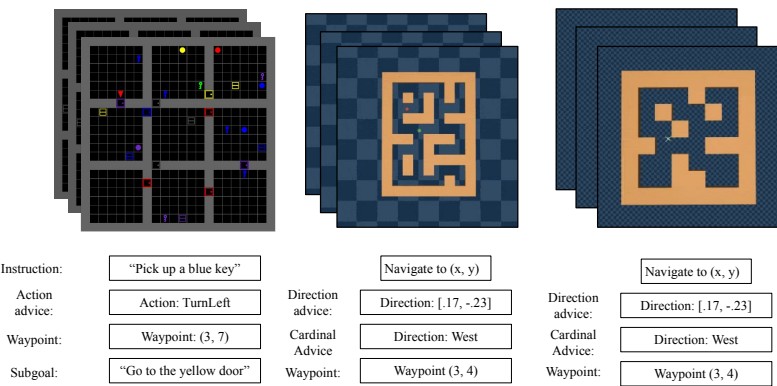

Figure 3: Evaluation Domains. (Left) BabyAI (Middle) Point Maze Navigation (Right) Ant Navigation. The associated task instructions are shown, as well as the types of advice available in each domain.

**BabyAI:** In the open-source BabyAI [8] grid-world, an agent is given tasks involving navigation, pick and place, door-opening and multi-step manipulation. We provide three types of advice:

1. **Action Advice:** Direct supervision of the next action to take.
2. **OffsetWaypoint Advice:** A tuple (x, y, b), where (x, y) is the goal coordinate minus the agent's current position, and b tells the agent whether to interact with an object.
3. **Subgoal Advice:** A language subgoal such as "Open the blue door."

**2-D Maze Navigation (PM):** In the 2D navigation environment, the goal is to reach a random target within a procedurally generated maze. We provide the agent different types of advice:

1. **Direction Advice**: The vector direction the agent should head in.
2. **Cardinal Advice**: Which of the cardinal directions (N, S, E, W) the agent should head in.
3. **Waypoint Advice**: The (x,y) position of a coordinate along the agent's route.
4. **OffsetWaypoint Advice**: The (x,y) waypoint minus the agent's current position.

**Ant-Maze Navigation (Ant):** The open-source ant-maze navigation domain [14] replaces the simple point mass agent with a quadrupedal "ant" robot. The forms of advice are the same as the ones described above for the point navigation domain.

In all domains, we describe advice forms provided each timestep (Action Advice and Direction Advice) as "low-level" advice, and advice provided less frequently as "high-level" advice. We present experiments involving both scripted coaches and real human-in-the-loop advice.

## 4.2 Experimental Setup

For the environments listed above, we evaluate the ability of the agent to perform grounding efficiently on a set of training tasks, to learn new test tasks quickly via advice distillation, and to leverage one form of advice to bootstrap another. The details of the exact set of training and testing tasks, as well as architecture and algorithmic details, are provided in the appendix.

We evaluate all the environments using the metric of **advice efficiency** rather than sample efficiency. By advice efficiency, we are evaluating the number of instances of coach-in-the-loop advice that are needed in order to learn a task. In real-world learning tasks, this coach is typically a human, and the cost of training largely comes from the provision of supervision (rather than time the agent spends interacting with the environment). The same is true for other forms of supervision such as behavioral cloning and RL (unless the human spends extensive time instrumenting the environment to allow autonomous rewards and resets). This "advice units" metric more accurately reflects the true quantity we would like to measure: the amount of human time and effort needed to provide a particular course of coaching. For simplicity, we consider every time a supervisor provides any supervision, such as a piece of advice or a scalar reward, to constitute one **advice unit**. We measure efficiency in terms of how many advice units are needed to learn a task. We emphasize that this metric makes a strong simplifying assumption—that all forms of advice have the same cost—which is certainly not true for real-world supervision. However, it is challenging to design a metric which accurately captures human effort. In Section 4.7 we validate our method by measuring the *real human interaction time* needed to train agents. We also plot more traditional sample efficiency measures in Appendix D.

We compare our proposed framework to an RL baseline that is provided with a task instruction but no advice. In the improvement phase, we also compare with behavioral cloning from an expert for environments where it is feasible to construct an oracle.

## 4.3 Grounding Prescriptive Advice during Training

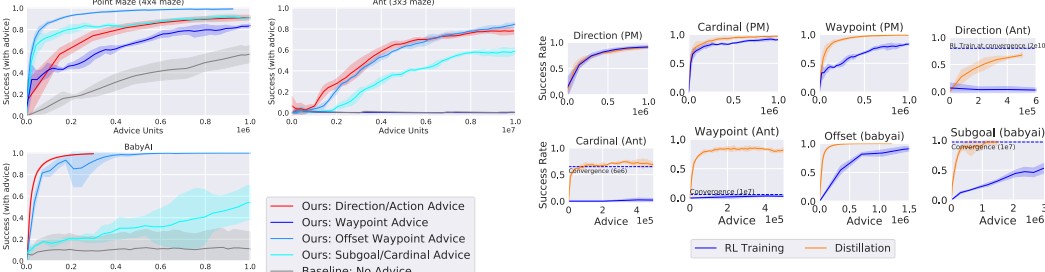

Figure 4: Left: Performance during the grounding phase (Section 3.2). All curves are trained with shaped-reward RL. We compare agents which condition on high-level advice (shades of blue) to ones with access to low-level advice (red) to an advice-free baseline (gray). Takeaways: (a) the agent is able to ground advice, which suggests that our advice-conditional policy may be useful for coaching; (b) grounding certain high-level advice forms through RL is slow, which is why bootstrapping is necessary. Right: Bootstrapping is able to quickly use existing grounded advice forms (OffsetWaypoint for Point Maze and Ant Maze envs, ActionAdvice for BabyAI) to ground additional forms of advice.

Fig 4 shows the results of the grounding phase, where the agent grounds advice by training an advice-conditional policy through RL. We observe the the agent learns the task more quickly when provided with advice, indicating that the agent is learning to interpret advice to complete tasks. However, we also see that the agent fails to improve much when conditioning on some more abstract forms of advice, such as waypoint advice in the ant environment. This indicates that the advice form has not been grounded properly through RL. In cases like this, we instead must instead ground these advice forms through bootstrapping, as discussed in Section 3.2.

## 4.4 Bootstrapping Multi-Level Feedback

Once we have successfully grounded the easiest form of advice, in each environment, we efficiently ground the other forms using the bootstrapping procedure from Section 3.2. As we see in Fig 4, bootstrap distillation is able to ground new forms of advice significantly more efficiently than if we start grounding things from scratch with naïve RL. It performs exceptionally well even for advice forms where naïve RL does not succeed at all, while providing additional speed up for environments

where it does. This suggests that advice is not just a tool to solve new tasks, but also a tool for grounding more complex forms of communication for the agent.

## 4.5 Learning New Tasks with Grounded Prescriptive Advice

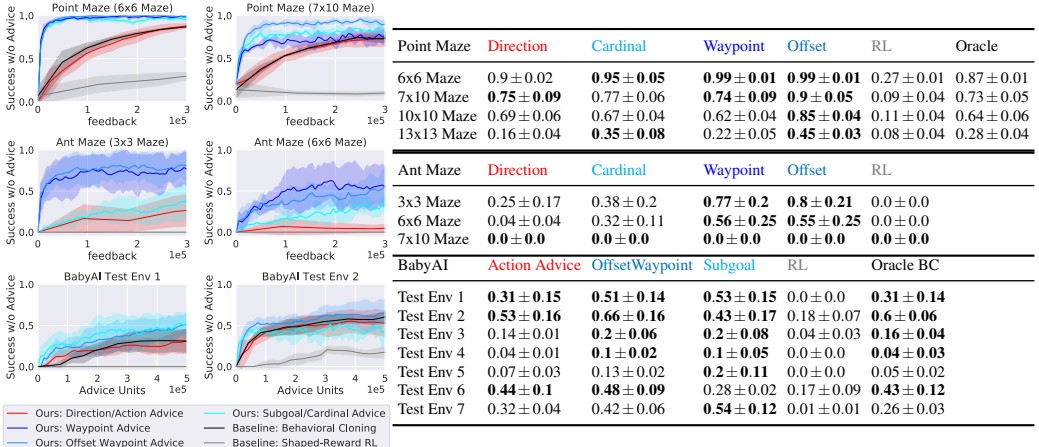

| Point Maze | Direction | Cardinal | Waypoint | Offset | RL | Oracle |
|---|---|---|---|---|---|---|
| 6x6 Maze | 0.9±0.02 | **0.95±0.05** | **0.99±0.01** | **0.99±0.01** | 0.27±0.01 | 0.87±0.01 |
| 7x10 Maze | **0.75±0.09** | 0.77±0.06 | **0.74±0.09** | **0.9±0.05** | 0.09±0.04 | 0.73±0.05 |
| 10x10 Maze | 0.69±0.06 | 0.67±0.04 | 0.62±0.04 | **0.85±0.04** | 0.11±0.04 | 0.64±0.06 |
| 13x13 Maze | 0.16±0.04 | **0.35±0.08** | 0.22±0.05 | **0.45±0.03** | 0.08±0.04 | 0.28±0.04 |

| Ant Maze | Direction | Cardinal | Waypoint | Offset | RL | |
|---|---|---|---|---|---|---|
| 3x3 Maze | 0.25±0.17 | 0.38±0.2 | **0.77±0.2** | **0.8±0.21** | 0.0±0.0 | |
| 6x6 Maze | 0.04±0.04 | 0.32±0.11 | **0.56±0.25** | **0.55±0.25** | 0.0±0.0 | |
| 7x10 Maze | **0.0±0.0** | **0.0±0.0** | **0.0±0.0** | **0.0±0.0** | 0.0±0.0 | |

| BabyAI | Action Advice | OffsetWaypoint | Subgoal | RL | Oracle BC |
|---|---|---|---|---|---|
| Test Env 1 | **0.31±0.15** | **0.51±0.14** | **0.53±0.15** | 0.0±0.0 | **0.31±0.14** |
| Test Env 2 | **0.53±0.16** | **0.66±0.16** | **0.43±0.17** | 0.18±0.07 | **0.6±0.06** |
| Test Env 3 | 0.14±0.01 | **0.2±0.06** | **0.2±0.08** | 0.04±0.03 | **0.16±0.04** |
| Test Env 4 | 0.04±0.01 | **0.1±0.02** | **0.1±0.05** | 0.0±0.0 | **0.04±0.03** |
| Test Env 5 | 0.07±0.03 | 0.13±0.02 | **0.2±0.11** | 0.0±0.0 | 0.05±0.02 |
| Test Env 6 | **0.44±0.1** | **0.48±0.09** | 0.28±0.02 | 0.17±0.09 | **0.43±0.12** |
| Test Env 7 | 0.32±0.04 | 0.42±0.06 | **0.54±0.12** | 0.01±0.01 | 0.26±0.03 |

Figure 5: Learning new tasks through distillation. The agent uses an already-grounded advice channel to perform the distillation process from Section 3.3 to train an advice-free agent. Results show the success rate of the advice-free new agent. Left, we show representative curves for a few environments. Colors designate supervision used: shades of blue = high level advice; red = low level advice; black = oracle demonstrations; gray = shaped rewards. Right: We show success rates (mean, std) over 3 seeds for a larger set of environments. Runs are bolded if std intervals overlapped with the highest mean. Success rates are evaluated at $3e5$ steps for Point Maze and Ant Maze and $5e5$ steps for BabyAI. Takeaway: once advice is grounded, in general it is most efficient to teach the agents new tasks by providing high-advice. There are occasional exceptions, discussed in Appendix G.

Finally, we evaluate whether we can use grounded advice to guide the agent through new tasks. In most cases, we directly used advice-conditional policies learned during grounding and bootstrapping. However, about half of the BabyAI high-level advice policies performed poorly on the test environments. In this case, we finetuned the policies with a few (<4k) samples collected with rollouts from a lower-level better grounded advice form.

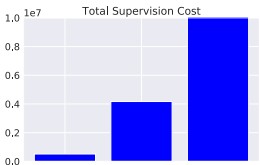

Figure 6: "Best advice" is OffsetAdvice. Y-axis includes advice from both grounding and improvement across all four Point Maze test envs. RL results stretch off the plot, indicating we were unable to run RL for long enough to converge to the success rates of the other methods.

As we can see in Fig 5, agents which are trained through distillation from an abstract coach on average train with less supervision than RL agents. Providing high-level advice can even sometimes outperform providing demonstrations, as the high-level advice allows the human to coach the agent through a successful trajectory without needing to provide an action at each timestep. It is about as efficient to provide low-level advice as to provide demos (when demos are available), as both involve providing one supervision unit per timestep.

Advice grounding on the new tasks is not always perfect, however. For Instance, in BabyAI Test Env 2 in Figure 5, occasional errors in the advice-conditional policy's interpretation of high-advice result in it being just as efficient efficient to provide low-level advice or demos as it is to provide high-level advice (though both are more efficient than RL). When grounding is poor, the converged final policy may not be fully successful. Baseline methods, in contrast, may ultimately converge to higher rates, even if they take far more samples. For instance, RL never succeeds in AntMaze 3x3 and 6x6 in the plots in Figure 5, but if training is continued for 1e6 advice units, RL achieves near-perfect performance, whereas our method plateaus. This suggests our method is most useful when costly supervision is the main constraint.

The curve in Figure 5 is not entirely a fair comparison - after all, we are not taking into account the advice units used to train the advice-conditional surrogate policy. However, it's also not fair to include this cost for each test env, since the up-front cost of grounding advice gets amortized over a large set of downstream tasks. Figure 6 summarizes the total number of samples needed to train each model to convergence on the Point Maze test environments, including all supervision provided during grounding and improvement. We see that when we use the best advice form, our method is 8x more efficient than demos, and over 20x more efficient than dense-reward RL. In the PointMaze environment, the cost of grounding becomes worthwhile with only 4 test envs. In other environments such as Ant, it may take many more test envs than the three we tested on. This suggests that our method is most appropriate when the agent will be used on a large set of downstream tasks.

## 4.6 Off-Policy Advice Relabeling

One limitation of the improvement phase as described Section 4.5 is that the human coach has to be continuously present as the agent is training to provide advice on every trajectory. We relax this requirement by providing the advice in hindsight rather than in-the-loop using the procedure from Section 3.3. Results are shown in Figure 7. IN the Point Maze and Ang envs, this DAgger-like scheme for soliciting advice performs greater than or equal to real-time advice. However, it performs worse in the BabyAI environment. In future work we will explore this approach further, as it removes the need for a human to be constantly present in the loop and opens avenues for using active learning techniques to label only the most informative trajectories.

## 4.7 Real Human Experiments

To validate the automated evaluation above (and determine whether our "advice unit" metric is a good proxy for human effort), we performed an additional set of experiments with human-in-the-loop coaches. Advice-conditional surrogate policies were pre-trained to follow advice using a scripted coach. The coaches (all researchers at U.C. Berkeley) then coached these agents through solving new, more complex test environments. Afterwards, an an advice-free policy was distilled from the successful trajectories. Humans provided advice

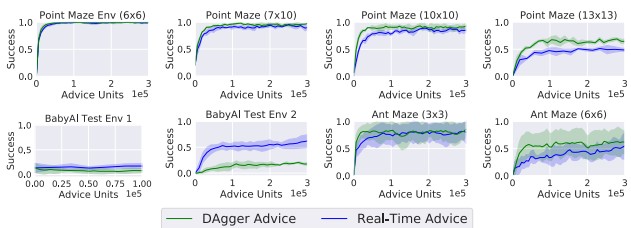

Figure 7: All curves show the success rate of an advice-free policy trained via distillation from an advice-conditional surrogate policy. All curves use the OffsetWaypoint advice form, and results are averaged over three seeds. Takeaway: DAgger performs well on some environments (Point Maze, Ant) but poorly on others (BabyAI).

through a click interface. (For instance, they could click on the screen to provide a.) See Fig 8.

In the BabyAI environment, we provide OffsetWaypoint advice and compare against a behavioral cloning (BC) baseline where the human provided per-timestep demonstrations using arrow keys. Our method's is higher variance and has a slightly lower mean success rate, but results are still largely consistent with Figure 5, which showed that for the BabyAI env BC is competitive with our method.

In the Ant environment, demonstrations aren't possible, and the agent does not explore well enough to learn from sparse rewards. We compare against the performance of an agent coached by a scripted coach providing dense, shaped rewards. We see that the agent trained with 30 minutes of coaching by humans performs comparably to an RL agent trained with 3k more advice units.

## 5 Related Work

The learning problem studied in this paper belongs to a more general class of human-in-the-loop RL problems [1, 25, 30, 47, 12]. Existing frameworks like TAMER [25, 45] and COACH [30, 4] also use interactive feedback to train policies, but are restricted to scalar or binary rewards. In contrast, our work formalizes the problem of learning from arbitrarily complex feedback signals. A distinct line of work looks to learn how to perform tasks from binary feedback with human preferences, for example by indicating which of two trajectory snippets a human might prefer [10, 21, 47, 27].

These techniques receive only a single bit of information with every human interaction, making human supervision time-consuming and tedious. In contrast, the learning algorithm we describe uses higher-bandwidth feedback signals like language-based subgoals and directional nudges, provided sparsely, to reduce the required effort from a supervisor.

Learning from feedback, especially provided in the form of natural language, is closely related to instruction following in natural language processing [7, 3, 32, 41]. In instruction following problems, the goal is to produce an *instruction-conditional* policy that can generalize to new natural language specifications of behavior (at the level of either goals or action sequences [24] and held-out environments. Here, our goal is to produce an *unconditional* policy that achieves good task success autonomously—we use instruction following models to interpret interactive feedback and scaffold the learning of these autonomous policies. Moreover, the advice provided is not limited to task-level specifications, but instead allows for real-time, local guidance of behavior. This provides significantly greater flexibility in altering agent behavior.

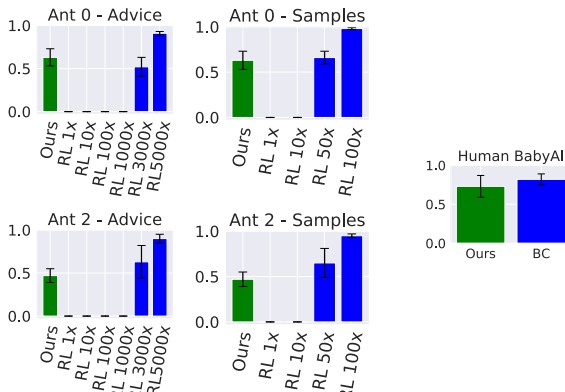

Figure 8: Left, Middle: We compare the success of an advice-free policy trained in two test envs with real human coaching to a RL policy trained with a scripted reward. "RL 10x" means RL policy received 10x more advice units (left) or samples (middle). Right: success of advice-free policies trained with 30 mins of human time. Humans either coach the agent with our method or provide demos. Sample sizes are n=2 per condition for Ant, n=3 per condition for BabyAI, so the results are suggestive not conclusive.

The use of language at training time to scaffold learning has been studied in several more specific settings [29]: Co-Reyes et al. [11] describe a procedure for learning to execute fixed target trajectories via interactive corrections, Andreas et al. [2] use language to produce policy *representations* useful for reinforcement learning, while Jiang et al. [22] and Hu et al. [18] use language to guide the learning of hierarchical policies. Eisenstein et al. [13] and Narasimhan et al. [35] use side information from language to communicate information about environment dynamics rather than high-value action sequences. In contrast to these settings, we aim to use interactive human in the loop advice to learn policies that can autonomously perform novel tasks, even when a human supervisor is not present.

## 6 Discussion

**Summary:** In this work, we introduced a new paradigm for teacher in the loop RL, which we refer to as coaching augmented MDPs. We show that CAMPDs cover a wide range of scenarios and introduce a novel framework to learn how to interpret and utilize advice in CAMDPs. We show that doing so has the dual benefits of being able to learn new tasks more efficiently in terms of human effort *and* being able to bootstrap one form of advice off of another for more efficient grounding.

**Limitations:** Our method relies on accurate grounding of advice, which does not always happen in the presence of other correlated environment features (e.g. the advice to "open the door," and the presence of a door in front of the agent). Furthermore, while our method is more efficient than BC or RL, it still requires significant human effort. These limitations are discussed further in Appendix G.

**Societal impacts:** As human in the loop systems such as the one described here are scaled up to real homes, privacy becomes a major concern. If we have learning systems operating around humans, sharing data and incorporating human feedback into their learning processes, they need to be careful about not divulging private information. Moreover, human in the loop systems are constantly operating around humans and need to be especially safe.

**Acknowledgments:** Thanks to experiment volunteers Yuqing Du, Kimin Lee, Anika Ramachandran, Philippe Hansen-Estruch, Alejandro Escontrela, Michael Chang, Sam Toyer, Ajay Jain, Dhruv Shah, Homer Walke. Funding by NSF GRFP and DARPA's XAI, LwLL, and/or SemaFor program, as well as BAIR's industrial alliance programs.

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
