# A Plots

Unless otherwise noted, all graphs are averaged over 3 seeds and show std error bars. All tables show mean and std errors.

# B Environments

In all environments, at each timestep the agent's policy conditions on the last unit of advice which the coach provided.

## B.1 D4RL Point Maze

This environment is a modified version of the environment found in the D4rl benchmark [14]. The state space consists of the agent's position and velocity, the target position, and a representation of the maze configuration.

The scripted coach is derived from the waypoint controller provided with the D4rl codebase. The waypoint controller finds a sequence of waypoints tracing the shortest path to the goal and computes the optimal direction the agent should head next, taking into account the next waypoint and the agent's current velocity. From this waypoint controller, we compute four advice types:

1. Direction - Optimal x-y direction to head in according to the waypoint controller.
2. Cardinal - One-hot encoding of whichever cardinal direction (N, S, E, W) has the greatest vector dot product with the optimal direction.
3. Waypoint - X-Y position of the next waypoint according to the waypoint controller.
4. OffsetWaypoint - Difference between the x-y position of the next waypoint according to the waypoint controller and the agent's current position.

Modifications from the original environment include:

1. Each reset, randomize the position of the agent's position and the goal. During training (but not test time) we also randomize maze wall configurations.
2. Modify the observation space to consist of the agent's position and velocity, the goal position, and a symbolic representation of the agent's grid. The grid is flattened and concatenated with the rest of the observation.
3. Custom semi-sparse reward provided to the agent every time it achieves an additional waypoint on the optimal path to goal.
4. Frame skip of 2.

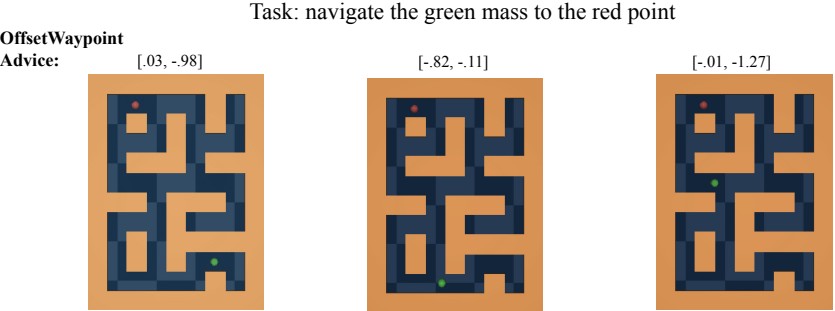

Task: navigate the green mass to the red point

OffsetWaypoint
Advice:  [.03, -.98]  [-.82, -.11]  [-.01, -1.27]

Figure 9: Example of advice offered during a trajectory in the Point Maze domain with OffsetWaypoint hints.

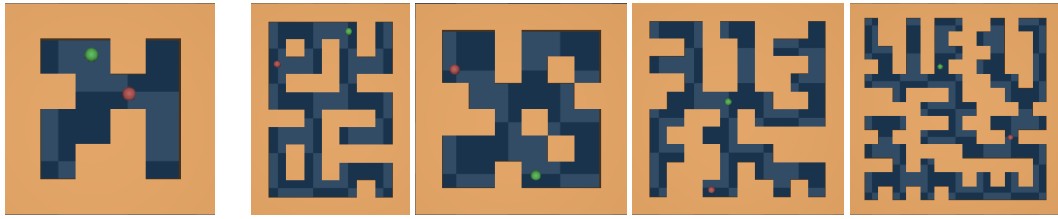

Figure 10: Left: The Point Maze grounding environment consists of randomized grids of this size. Tasks involve navigating to a particular position in the maze.

## B.2 Ant

This environment is a modified version of the environment found in the D4rl benchmark [14]. The agent's state space consists of the position and velocity of each of its joints, the target position, and a representation of the maze configuration. The advice forms used are identical to those in the Point Maze environment. Modifications include:

1. Change the gear ratio of the ant's legs to 30.

2. Modify the observation space to consist of the agent's position, goal position, the positions and velocities of each joint, and a symbolic representation of the agent's grid.

3. Implement a custom shaped reward. The reward is the dot product between two normalized vectors: the direction the agent's torso traveled in the last timestep, and the optimal direction for the torso to travel according to the environment waypoint controller. This reward was inspired by [17]. The agent is given an additional semi-sparse reward whenever it achieves a waypoint specified by the waypoint controller.

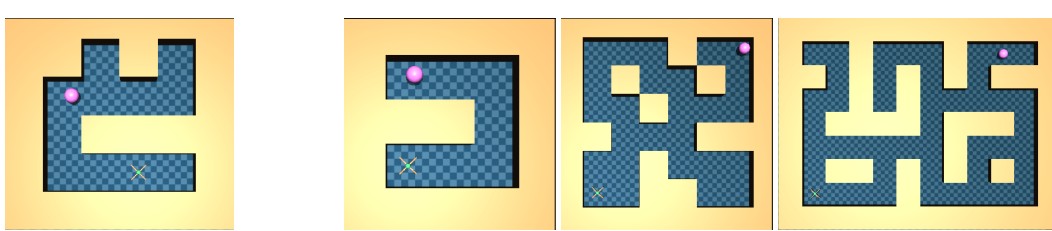

Figure 11: Left: The Ant grounding environment consists of randomized grids of this size. Tasks involve navigating to a particular position in the maze.

## B.3 BabyAI

This environment is a modified version of the environment found in the BabyAI benchmark [8]. Modifications include:

1. Make the environment fully observable.

2. Modify the observation space to be egocentric. The observation is rotated and placed within a larger padded grid such that the agent appears at the same coordinate at all times.

3. Define a few custom tasks.

We use three advice types in this environment:

1. Action Advice - One-hot encoded vector specifying which discrete action to take next.

2. OffsetWaypoint Advice - X-Y coordinate offset of the location it should reach in $k$ timesteps, where $k \sim U[2, 20]$. After $k$ timesteps, another waypoint is sampled. The agent also receives a boolean token indicating whether it interacts with an object while reaching this waypoint. The agent also sees how many timesteps ago the advice was given.

3. Subgoal Advice - the agent is given a scripted language subgoal such as "Open the red door" or "Pick up the green key at [6, 3]".

We train and test on multiple levels, all of which are procedurally generated each reset. All grids except PutNextLocal are 22x22 grids which look similar to those shown in Fig 12.

**Training Envs:**

1. GoTo: find an object, e.g. "Go to a purple ball"
2. Open: open a particular door, e.g. "Open the gray door"
3. PickUp: pick up a particular object, e.g. "Pick up a green box"
4. PutNext: put one object adjacent to another, e.g. "Put a red ball next to a blue box"
5. PutNextLocal: Like PutNext, but grid is 8x8

**Testing Envs:**

1. GoToDistractors: like GoTo, but the maze 60 distractor objects rather than 18
2. GoToYellow: like GoTo, but the target is always yellow, which was never the case during training
3. PutNextSame: put an object adjacent to a matching one, e.g. "Put a green ball next to a key of the same color"
4. Unlock: like Open, but door is locked and can only be opened when agent is holding a key of the same color
5. GoToDistractorsFixed: like GoToDistractors, but target object is always a red ball
6. PutNextSame: like PutNextSame, but target object is always a red ball
7. Unlock: like Unlock, but target object is always a red ball

Task: put a purple ball next to a blue key

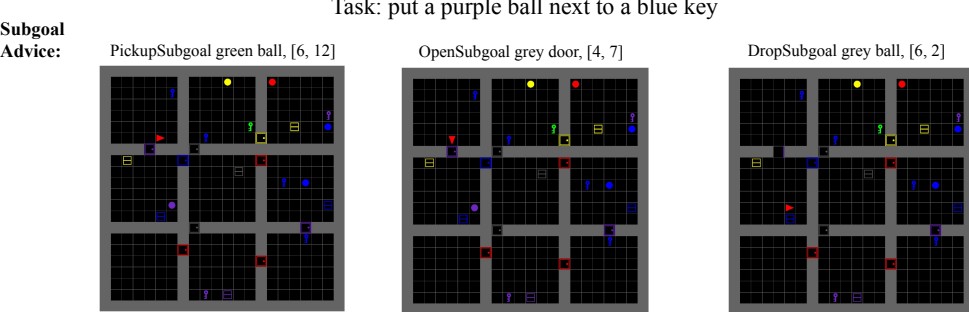

Figure 12: Example of advice offered during a trajectory in the BabyAI domain with Subgoal advice.

## C   Code

Code can be found at https://github.com/rll-research/teachable under the MIT licence. The codebase incorporates elements of the meta-mb codebase, found at https://github.com/iclavera/meta-mb under the MIT license, the BabyAI codebase, found at https://github.com/mila-iqia/babyai under the BSD-3-Clause license, the d4rl codebase, found at https://github.com/rail-berkeley/d4rl under the Apache licence, and https://github.com/denisyarats/pytorch_sac which uses the MIT License.

## D   Sample Efficiency

Here, we report the same curves as shown in Figures 4 and 5, but here we show **samples** on the x-axis rather than advice units. Takeaways include:

1. While low-level advice is less advice-efficient, its sample efficiency is equal or better than high-level advice. This makes sense, since the same feature which makes high-level advice advice-efficient - infrequent provision - also makes it more challenging to interpret than low-level advice.

2. RL still performs poorly in some environments (e.g. the BabyAI and Point Maze envs in Fig 15, but in environments where distillation using advice doesn't work very well, such as Ant Maze, RL starts with worse performance but ultimately converges higher.

3. Bootstrapping is typically more sample-efficient than RL training.

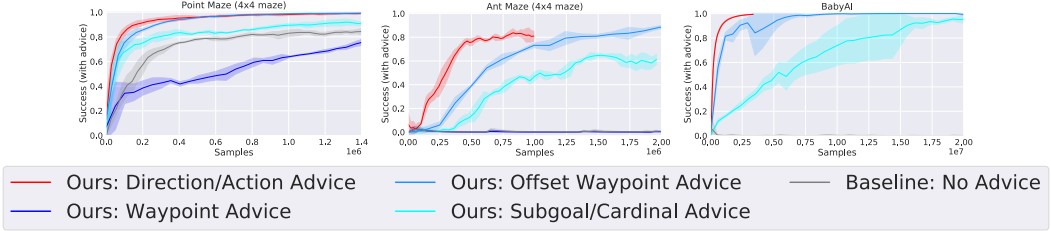

Figure 13: This shows **sample efficiency** in the grounding phase, similar to the advice-efficiency plot in Fig 4.

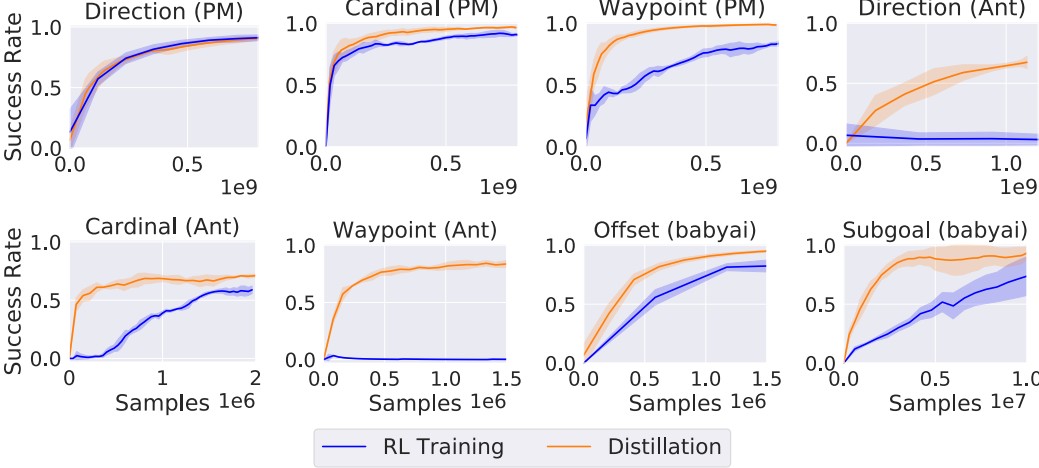

Figure 14: This shows **sample efficiency** during bootstrapping, similar to the advice efficiency plot in Fig 4

# E  Human Experiment Details

**Instructions**: Participants were told the goal of the environment they were asked to coach an agent in, and the advice/supervision interface was explained to them. Paricipants were allowed to practice in the environment until they reported to us that they felt comfortable with the controls. We (the authors) answered any questions they had about the task during this process. Participants collected data for 30 consecutive minutes.

# F  Algorithm and Architecture

## F.1  Algorithm

We train our our surrogate policy using using PPO as implemented in [19]. During distillation, we use behavioral cloning. Our codebase is based upon the imitation learning code from [8], with modifications to sample timesteps individually rather than as full trajectories.

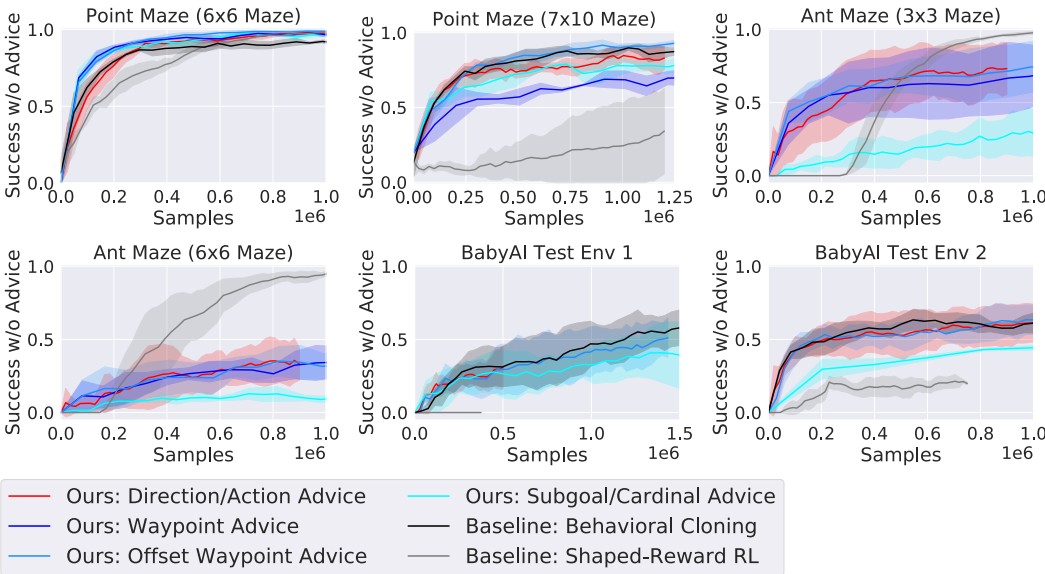

Figure 15: This shows **sample efficiency** in the improvement phase, similar to the advice-efficiency plot in Fig 5.

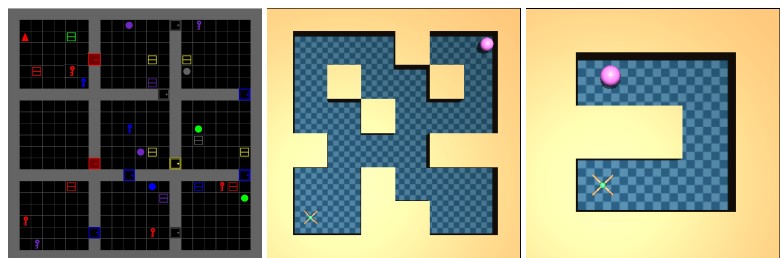

Figure 16: Environments used in human experiments. Left: the agent's task is to open a locked door, which involves first picking up a matching key. To speed up training, the agent, key, and target door are all spawned in the same room, and the key is always at the same location. The agent had never seen a locked door during training. Middle, Right: These are the 3x3 and 6x6 mazes used in the scripted experiments.

## F.2    Model

We build upon the architecture provided along with the BabyAI environment [19], shown with modifications in Figure 17. Modifications include swappint the LSTM for the actor-critic model here `https://github.com/denisyarats/pytorch_sac` and incorporating advice.

## F.3    Hyperparameters

We chose model hyperparameters by sequentially sweeping over learning rate, batch size, control penalty coefficient, entropy coefficient, discount, and update steps per iteration.

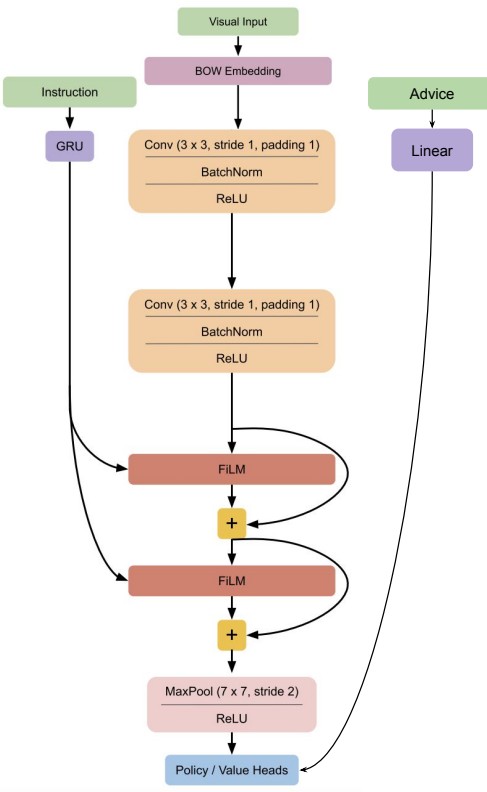

Figure 17: Architecture diagram modified from BabyAI 1.1 [19]. For the Point Maze and Ant envs, which do not have image input or instructions, the advice is linearly embedded, concatenated with the state, and passed to MLP actor and critic models.

| Hyperparameter | Value |
|---|---|
| **Optimizer** | |
| Adam $\beta_1$ | .9 |
| Adam $\beta_2$ | .999 |
| Adam $\varepsilon$ | 1e-5 |
| Adam lr | 1e-3 |
| **Model architecture** | |
| Advice embedding dim | 128 |
| Hidden layer size (actor & critic) | 128 |
| # hidden layers (actor & critic) | 2 |
| Activations | Tanh |
| **PPO** | |
| Env steps per update | 800 |
| Entropy bonus coef | 1e-3 |
| Discount | .99 |
| Value loss coef | .5 |
| Clip $\varepsilon$ | .2 |
| Grad clip | .5 |
| Batch size | 512 |
| GAE $\lambda$ | .95 |
| Update steps | 20 |
| **Distillation** | |
| Batch size | 512 |
| Env steps per collection round | 800 |
| Distillation steps per collection round | 100 |
| Buffer capacity | 1M |

In addition, there were some environment-specific hyperparameters:

- BabyAI used 15 distillation steps per collection round, 128-dimensional embeddings for the input elements (see Fig 17 for full input embedding). PPO took 750 env steps per update, but this was modified for memory capacity reasons not performance.
    - 15 embedding steps per collection round
    - input embedding as in Fig 17 with 128-dim embeddings
    - 1000 env steps per update for RL, 750 for distillation
    - discount 0 for Action Advice, .25 for others
    - 10 update steps
    - RL baseline was tuned separately and used lr 1e-4 and entropy coef .005
- Point Maze used a control penalty coef of .01.
- Ant Maze used a control penalty coef of .1.

## G  Failure Cases and Challenges

Cases in which our proposed method fails can be broken into 2 categories:

1. **Advice is not be grounded correctly.** We encountered this often. For instance, the poor performance on 13x13 Ant Maze in diagram 5 was largely due to the fact that even with advice, the agent typically failed at the task. Strategies for addressing this include (1) finetuning the surrogate policy for a few iterations in the test env, either through RL or through bootstrapping from a more-successful lower-level advice form, and (2) swapping between advice forms, so that if an agent receiving high-level advice (e.g. subgoals) gets stuck in a particular state, the coach can switch to lower-level advice (e.g. cardinal direction to travel in) which is likely grounded better. Still, these methods are imperfect and imperfect grounding limits the agent's ability to generalize to arbitrary new tasks.

2. **Test-time tasks cannot be solved easily using previously-grounded advice.** Some test-time task might not be expressible in terms of high-level advice (e.g. subgoals) the agent understands. However, the agent can still be coached to success on this task using lower-level Action Advice. Future extensions to this work will involve providing abstract advice during easy portions of the task, but dropping down to lower-level advice during portions of the task where the agent isn't able to follow high-level advice.

There are also additional challenges with using our method:

1. **Advice representation choice matters.** To effectively learn from advice, the advice must be represented in a way which is easy for the model to interpret. For example, Figure 4 shows that the agent learns far more easily with OffsetWaypoint than Waypoint coaching despite the fact that both contain the same information (the agent can compute an OffsetWaypoint by subtracting the Waypoint from its current position). Using this method may require some engineering effort to choose an appropriate advice representation which is grounded quickly and generalizes well to the test environments.

2. **Our approach still requires substantial human effort.** In future work, we plan to reduce the amount of human supervision through strategies including (a) pretraining with an unsupervised skill discovery, (b) moving almost entirely to off-policy advice provision, (c) only providing advice on key trajectories where the agent is uncertain, and (d) interspersing human advice with periods of unsupervised goal-reaching practice.

3. **There is no scalable, reliable evaluation metric.** As mentioned previously, our "Advice Units" metric assumes a unit from each advice form is equally costly, which is clearly not true - for instance, binary rewards and waypoints can be provided quickly, language advice takes a bit longer, and dense shaped rewards may be difficult for a human to provide accurately even with plenty of time. Our real-human experiments provide a more reliable comparison, but human evaluations aren't scalable. They also suffer from comparison challenges: for instance, several participants complained about challenges adjusting to the click-and-scroll interface used to provide advice during our human experiments, whereas participants who

provided demos through the arrow-key baseline were more familiar. Finally, our comparison with RL assumes that a human is providing reward each time. However, alternative training setups involve humans spending a lot of up-front time to build a simulator or to instrument the real world with rewards and resets, then having the agent practice autonomously. We don't have a clear way to compare against this approach.

4. **Hyperparameters and implementation choices make cross-method comparisons challenging** We found that the design choices needed to achieve good performance differed across conditions. For instance, for our method we needed to choose an appropriate advice representation, for behavioral cloning we had to determine what degree of noise to add to achieve optimal performance, and for RL we had to experiment with different reward functions. We made a good-faith effort to tune all conditions, but it is still unclear exactly what constitutes a fair comparison.

## H Compute

The experiments in this paper were run on 8 11019MiB GPUs for about 3 weeks.

## I Robustness to Noise

Unlike in the real-human experiments, where advice provided is often noisy, the advice in most of our simulated experiments never makes mistakes. We evaluated whether our method can be used to coach agents when the simulated advice is noisy by introducing noise into the improvement phase. OffsetWaypoint advice was provided by a scripted coach which gridifies the maze and plots a path through the grid. The noise introduced randomly replaced a certain fraction of waypoints with points sampled from adjacent grid cells. Incorrect waypoints were provided for as many timesteps as it takes the agent to reach the next waypoint. Results are reported in Figure 18. Heavy levels of noise significantly hurt agent performance, although this is perhaps an unrealistically bad noise model (for instance, a real teacher would likely recognize that the agent is failing to achieve a particular waypoint and correct the error).

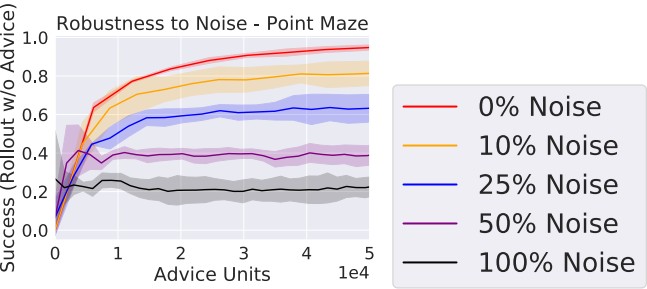

Figure 18: Performance on the improvement phase in the Point Maze environment. The noise percentage refers to the percent of waypoints which were replaced by adjacent incorrect waypoints.

## J Alternative Ways to Use Advice

We explored several alternative ways to provide advice, but ultimately found the approach presented in our works most reliably.

### J.1 Advice Reconstruction

Rather than providing advice as an input to the agent's observation, we incorporated advice by adding an auxiliary loss to predict it, similar to [48]. While we found this improved performance slightly over a pure RL baseline, we found the advice wasn't competitive with our approach and wasn't able to speed up learning in challenging environments like Ant.

## J.2 Hindsight Relabeling

Rather than provide prescriptive advice, we explored having the coach provide advice by relabeling an agent's trajectory with the goal it achieved. We can then train this now-successful relabeled trajectory using supervised learning, as was done in [36]. However, we found that hindsight relabeling performed poorly, except on the simplest environments. However, we only tried a very naive approach to getting this method to work, and it's possible more sophisticated methods could succeed here.

## J.3 Hierarchical RL

We explored an alternate method of using advice with hierarchical rl. We modified the grounding phase to train an advice-conditional surrogate policy $q_\phi(a|s, \tau, \underline{\mathbf{c}})$ as described in Section 3, but also do supervised training of an advice generation high-level policy $h_\psi(\underline{\mathbf{c}}|s, \tau)$ which predicts advice. During the improvement phase, the coach directly provides advice to the low-level policy to coach the agent to success on the new task. Simultaneously, we can fine-tune the high-level policy on advice from this environment. (No rewards or low-level supervision is provided during this phase.) Unlike in our main method, we do not learn an advice-free policy $\pi(a|s, \tau)$. At evaluation time, $h_\psi(\underline{\mathbf{c}}|s, \tau)$ generates advice, which $q_\phi(a|s, \tau, \underline{\mathbf{c}})$ executes. Results using this approach are shown in Fig 19, where we see that it performs comparably to our approach (labeled "Distill Flat") across a range of advice types and conditions. However, we only show results on a few simple environments and advice types. With more complex advice representations (e.g. waypoints, subgoals), we found we were not able to even learn a low-level policy which could predict advice well enough to succeed on the train levels, much less on the test environments reported in Fig 19.

| Env | Advice | Distill Flat | Finetune Hierarchical |
|---|---|---|---|
| PointMaze 6x6 | Direction | $0.98 \pm 0.01$ | $0.99 \pm 0.0$ |
| PointMaze 6x6 | Cardinal | $0.34 \pm 0.39$ | $0.27 \pm 0.32$ |
| PointMaze 7x10 | Direction | $0.91 \pm 0.03$ | $0.91 \pm 0.02$ |
| PointMaze 7x10 | Cardinal | $0.21 \pm 0.25$ | $0.21 \pm 0.25$ |
| PointMaze 7x10 | OffsetWaypoint | $0.97 \pm 0.02$ | $0.95 \pm 0.0$ |
| PointMaze 10x10 | Direction | $0.84 \pm 0.05$ | $0.94 \pm 0.04$ |
| PointMaze 10x10 | Cardinal | $.2 \pm 0.24$ | $0.17 \pm 0.21$ |
| PointMaze 10x10 | OffsetWaypoint | $0.96 \pm 0.02$ | $0.94 \pm 0.04$ |

Figure 19: Success rate of the distillation phase using our method vs the hierarchical RL method. Typically, these methods perform at approximately the same rate. However, these test environment evaluations were only done for advice forms where the agent was able to learn a decent advice predictor on the train environments in the first place. (Note: this experiment was run on an earlier iteration of the codebase and therefore results aren't directly comparable to Fig 5).