# OpenReview forum: "Teachable Reinforcement Learning via Advice Distillation"
_NeurIPS.cc/2021/Conference — NeurIPS 2021 Poster_

### Official Review · Reviewer_df4E · 2021-07-01

**Rating:** 6
**Confidence:** 4

**Summary:**

The authors propose a reinforcement learning approach where humans can
provide different forms of advice to complete a task. The proposed
approach, called CAMDPs, follows a two-stage process: Grounding and
Improvement.

During grounding the system learns how to interpret the advice. The
result of this phase is a surrogate policy. Grounding learns a mapping
from advice to actions using information from the true reward
function. Grounding can be seen as an ordinarily multitask RL, where
each task is a piece of advice, and the agent learns using normal RL
how to achieve the advice.

Distillation is used to learn a policy using supervised learning and
bootstrap distillation uses the previously learned policies for
individual advice to learn a policy for a particular task.

During improvement the coach provides advice to solve a new task. A very
similar approach is followed as bootstrap distillation, a new task
is solved using advice followed by a distillation process.

It is expected for the learned policies to be general enough to be
applied in other, although similar, domains.

The proposed system is evaluated in three, relatively similar,
domains. As expected, it is experimentally shown that the proposed
system learns faster with advice.


**Ethical Concerns:**

There are no ethical concerns

**Limitations And Societal Impact:**

Yes the authors adequately addressed the limitations and positive societal impact

**Main Review:**

Some positive aspects of this paper are that it can use different
types of advice and it is tested on different domains. It is also
positive that it can combine low-level advice and policies into
higher-level policies to solve more complex tasks.

On the negative side, the description of the method is not completely
clear and the proposed method seems to heavily depend on human
intervention in order to work. It also depends on some strong
assumptions in terms of how the advice is given, which makes it
difficult to assess how useful the proposed system will behave under
more realistic conditions.


It is not clear how the sub-goals are related to higher goals. It
seems to be that the user is responsible for providing this, but needs
clarification.

Also it is not clear how the pieces of advice are learned. Since more
abstract advice can be learned from more concrete advice, does it mean
that the system needs to first learn the less abstract advice (e.g.,
direction) before more abstract advises (e.g, cardinal and then
way-points) or can the advises be learned independently.

It is not clear how the proposed approach of combining low level
policies to learn more abstract policies compares to learning
options or other hierarchical learning approaches.

There are different types of advice (at different abstraction levels),
but in the evaluation phase they are treated equally. A more detailed
evaluation, considering these differences would certainly help to
understand the role of each type of advice.

At every state the teacher chooses the coaching function (if any) to
apply. This is currently hardwired, but it is not clear what was the
rationale to define this process (when to give advice and which type
of advice to use).

The authors should clarify the the types of advice is restricted to
those that suggest possible actions. For instance, a user may tell the
agent that what it is doing is wrong without providing any specific
hint of which action may be the right one.

It would be very helpful to give an example of one "typical" episode
with advice (e.g., when and what type of advice is given, how often, etc.)

It not clear how general is the approach. It is shown that the
learned advice can be used in larger versions of the same original
task. Can the advice be transferred between different tasks, e.g., from
BabyAI to Ant Navigation?

Also, the system assumes ideal conditions with perfect advice and no
noise. It is not clear how the proposed approach will behave under
more realistic conditions. It is easy to image scenarios where a user
gives wrong pieces of advises, with delays, or gives an advice that it
is too general from which the system is not able to learn, etc.

Small typos:
- a advice
- other We

----------
I have read the comments of the other reviewers and the authors' response.
I appreciate that the authors have performed further experiments to
strengthen their paper and I think that the paper could be publish,
however I still maintain my original acceptance score.

**Time Spent Reviewing:**

5

---

> ### Author Response · Authors · 2021-08-10
> **Authors' Response**
>
> Thank you for your helpful comments and suggestions! Please find responses to  individual concerns below:
>
> **“It also depends on some strong assumptions in terms of how the advice is given" >** Would the reviewer be able to clarify this further? We believe that the only assumptions on the advice are largely that it has to be informative about what actions to take in the future and provided to the agent in real-time (although we relax this second assumption here: https://tinyurl.com/exp3-dagger). Besides that, the advice can be in any form, without any particular restrictions. We would be glad to clarify this further.
>
> **“It is not clear how the sub-goals are related to higher goals.” >** We would like to note that the coaching is not limited to just “sub-goals”, which are just intermediate states, but can more broadly refer to advice forms like directional, waypoint, or even natural language advice. These pieces of advice are provided by a human teacher in the process of guiding an agent towards performing a new task, essentially involving a human supervisor taking a high level goal and breaking it down into intermediate advice pieces to guide an agent towards learning a new task.
>
> **“it is not clear how the pieces of advice are learned” >** Advice is grounded by training an advice-conditional surrogate policy on a set of smaller/simpler training environments. During the grounding phase, supervision is provided through rewards for RL. If the human has already taught the agent a lower-level form of advice, the RL training of the new advice form can be replaced by distillation from trajectories collected using the low-level advice surrogate policy.
>
> **“It is not clear how the proposed approach compares to learning options or other hierarchical learning approaches.” >** Most hierarchical RL (HRL) approaches operate in a somewhat different problem setting, where the high-level policy is learned without  human-in-the-loop supervision. However, our proposed approach is a natural complement to HRL because in addition to using advice to coach agents through new tasks, this advice could also be used as supervision for a high-level policy.  We are currently running experiments to show how our approach integrates easily into an HRL setup. See https://tinyurl.com/exp4-hrl for details.
>
> **“A more detailed evaluation, considering these differences would certainly help to understand the role of each type of advice” >** Great question! We did evaluate on the basis of advice efficiency because this is representative of the amount of effort a human supervisor would take to provide the agent with supervision. We also include an evaluation in terms of raw sample efficiency in Appendix D and find that while low-level advice is less advice-efficient, its sample efficiency is equal or better than high-level advice. Using advice of any form is more sample efficient than using a sparse-reward baseline. Dense reward baselines are competitive during RL training, but are less sample-efficient during distillation. Additionally, we ran human evaluation experiments where we allow humans to coach the agent towards learning new tasks and measure the time taken to actually provide coaching. This can be found at https://tinyurl.com/exp1-human and suggests that using advice, a human can coach an agent to high performance on a task more efficiently than with demonstrations or rewards.
>
> **“it is not clear what was the rationale to define this process (when to give advice and which type of advice to use).” >** Great question! In general, the question of when to provide advice is largely left up to the human supervisor. For certain types of advice like ActionAdvice, we can see that providing it at every time step is most effective since it provides action level direction. On other forms of advice like waypoint advice, only providing the advice when the previous waypoint has been accomplished is more effective. In general the rule of thumb here is that more abstract advice is provided less frequently.
>
> **“The authors should clarify the the types of advice is restricted to those that suggest possible actions” >** Indeed the types of advice that the agent is permitted to use have to be linked to what behavior should be performed in the future, and this framework is not able to handle “feedback” which is provided about behavior in the past. This would be very interesting future work!
>
> **“It would be very helpful to give an example of one "typical" episode with advice” >** One example of typical human-provided advice can be seen for the AntMaze here: https://tinyurl.com/exp1-human. Examples of advice in the other environments can be found in the Appendix A, specifically Figures 8 and 11.
>
> **“It not clear how general is the approach” >** As the experimental evaluation currently  stands, training is done in a single domain (like BabyAI or Ant) and the grounded behaviors are largely specific to that particular domain. But the exact same algorithm and system is applied to each of the domains. So the system itself is fully general, and easily applicable to any environment where it is feasible for a human to provide real-time coaching to a learning agent, although the current grounding will not transfer between BabyAI and ant navigation. However, if an agent has an expressive enough state and action space then the framework is general enough to actually operate across different domains as well.
>
> **“Also, the system assumes ideal conditions with perfect advice and no noise” >** To  help evaluate this further, we ran 2 sets of experiments (see shared response for details). Firstly, we added noise to advice during the improvement phase and found only slight performance drops even with 30% noisy advice. (https://tinyurl.com/exp2-noise). Secondly, we ran real human experiments and found that advice is more effective supervision than demos or rewards, even with inherently noisy human advice. (https://tinyurl.com/exp1-human)

---

> > ### Comment · Reviewer_df4E · 2021-08-11
> > **Clarification on one of my comments**
> >
> > > “It also depends on some strong assumptions in terms of how the advice is given" > Would the reviewer be able to   clarify this further? .... Besides that, the advice can be in any form, without any particular restrictions. We would be glad to clarify this further.
> >
> > This was a general comment related to the more specific comments given later in my review (for which you have already replied) that was made because:
> > - The original presented results were all performed in simulation and without noise
> > - The advices were given by a scripted teacher
> > - All advises were treated equally
> > With respect to your last comment "the advice can be given in any form, without any particular restrictions" that could be arguable, I have the impression that it needs to follow certain format/syntax and that it accepts only certain type of advice (I can think of other advices not applicable to your work, like, "great move" or "that's terrible" etc)

---

> > > ### Author Response · Authors · 2021-08-12
> > > **Further clarification**
> > >
> > > In response to your last bullet point: we assume that advice is prescriptive about what the agent should do next (so "great move" isn't applicable, as you said). We also do assume the agent knows the format of the teacher's advice input channel, but this channel can be very general and expressive (for instance, a string of text).

---

### Official Review · Reviewer_TZyF · 2021-07-16

**Rating:** 7
**Confidence:** 3

**Summary:**

The paper presents an interesting framework (CAMDPs) for teacher-in-the-loop-based RL, that takes into account rich interactive advice. The authors also show how the advice is interpreted in the Coaching augmented MDP. The framework comprises of 3 stages: grounding, improvement, and evaluation. In the grounding phase, the agent learns to interpret an 'advice', in the improvement phase the agent learns a policy using the advice and finally in the evaluation phase the learner deploys it's advice-independent policy. The claim is that the proposed pipeline helps a learner learn more efficiently, as it is able to account for richer forms of feedback. The method was tested on 3 rich RL environments: Baby AI, Ant Env, and Point Navigation Env, and evaluated based on its advice efficiency. The evaluations shpw that an agent learns quicker when presented with advice than when it is not.

**Limitations And Societal Impact:**

The paper addresses its limitations of assuming that the provided advice by the teacher is always optimal. They have left incorporating noisy advice for future work. However, this work is still far from deployment to be concerned about its negative societal impacts.

**Main Review:**

**Pros**

* The paper tackles an ambitious problem of learning from high-level advice in teacher-in-the-loop-based RL systems. The problem formulation was clean, and the three steps of the framework were well explained. The idea of using bootstrapping to interpret multi-level advice was particularly interesting.
* The experimental evaluation carried out on the rich RL environments was promising. The experimental analysis on the 3 RL environments was exhaustive.



**Cons**

* Evaluation: While the evaluation on the 3 rich RL environments was done in terms of the advice efficiency metric, it would be interesting to see how this method compares to the existing methods such as imitation learning, in terms of their sample efficiency. Furthermore, it would also be interesting to investigate the effect of richer/ more structured advice grammar and analyze further if there was a way to determine the optimal sequence of different advice types within a task, to improve performance fastest. It would be great to see the effect of this framework on a more real setting: perhaps on teaching an agent to solve mathematical equations, which comprises a more challenging advice set.



**General Comments**
This paper opens avenues for analysis of such rich feedback-based RL systems. It presents an interesting framework to analyze the richness of advice-based feedback w.r.t task. The different advice types investigated, and the framework presented for their distillation into a policy is novel. The evaluation of such a framework on the simulated RL environments looks promising, and there is scope to explore more interesting/complex advice grammar.


*Originality*: Novel Approach

*Clarity*: Largely clear.

*Quality*: Good

*Significance*: Important stepping stone to future work in this direction


**Time Spent Reviewing:**

4

---

> ### Author Response · Authors · 2021-08-10
> **Authors' Response**
>
> We thank you for your positive assessment of our work! We point you to the shared response for a broader discussion of new experiments we have run to address reviewer concerns. Please find responses to specific points below:
>
> **“it would be interesting to see how this method compares to the existing methods such as imitation learning, in terms of their sample efficiency” >** We have actually already run this comparison and reported results on the grounding phase in Appendix D in the supplementary material. Additional behavioral cloning comparisons, both with feedback efficiency and sample efficiency, have been added here: https://tinyurl.com/bc-curves. As can be seen, sample efficiency is improved in some cases but more generally sees less improvement than advice efficiency.
>
> **“if there was a way to determine the optimal sequence of different advice types within a task, to improve performance fastest” >** To understand this point, we conducted actual human in the loop experiments. In the BabyAI environment, we allow the human to switch between different forms of advice mid-trajectory. Empirically, we see that grounding of the lower-level advice on new environments is better than high-level advice (100% success, 92%+ per-timestep accuracy for Action Advice vs 98% success and 76% per-timestep accuracy for Offset Waypoint advice), but on the flip-side these lower level forms of advice are less advice efficient. Allowing humans to switch between advice forms lets them use the more efficient higher-level advice without getting stuck in the few situations where the agent interprets the advice incorrectly. We leave a more controlled study of the exact balance between different advice forms to future work, but note that the ability to use multiple different coaching forms to guide learning does greatly help performance and efficiency.
>
> **“Effect of this framework on a more real setting”:** While we are currently working on setting up a new domain as requested, we made the evaluation more realistic along the metric of realism by showing we can successfully replace the scripted teacher with human coaching and still outperform providing demonstrations, as shown here: https://tinyurl.com/exp1-human.

---

> > ### Comment · Reviewer_TZyF · 2021-09-02
> > **Queries sufficiently addressed**
> >
> > I am happy with the exhaustive results presented in the appendix, as well as the additional evaluations carried out. I am more inclined to accept the paper now.

---

### Official Review · Reviewer_NG65 · 2021-07-16

**Rating:** 6
**Confidence:** 3

**Summary:**

The paper proposes a mechanism Coaching Augmented MDP for agents to learn from interactive expert feedback to avoid assumptions on reward functions and agent infrastructure. Here, structured advice is provided by experts to guide the agent learning process. A policy learning algorithm is designed for interpreting and acting on the advice to solve a problem. Experiments are conducted on discrete and continuous control problems to show how such mechanism outperforms traditional RL models.

**Limitations And Societal Impact:**

The paper points out that the teacher in the loop is not a real-time human which can be noisy and compromised which are scenarios that need to be considered. The proposed method is a work towards having autonomous systems operate with rich interactive teacher feedback instead of just reward signals

**Main Review:**

Originality:
Though the proposed Coaching Augmented MDP is closely related to multi-task MDP and distillation, it still has novelty in processing more interactive feedback with sub-goals along with an un-conditional policy

Quality
The paper proposes a framework for multi-task MDP with teacher in the loop using different forms of coaching functions called advice. They also propose an algorithm to perform exploration using these coaching signals while the actual task can be performed without these signals. The stage-wise learning helps to achieve this. The technical details of  these method are explained in detail in the paper.
Experiments are conducted both in discrete and continuous control problems to verify that the proposed method outperforms traditional RL methods. The authors can clarify if the proposed mechanism is more time-consuming than the traditional RL techniques.

Clarity
The paper is well written and easy to follow.

Significance:
The work is significant to the community as it introduces the paradigm of teacher in the loop RL which considers interactive feedback during learning. The work looks like a promising step towards a broader class of feedback loop during RL training.


**Time Spent Reviewing:**

4

---

> ### Author Response · Authors · 2021-08-10
> **Authors' Response**
>
> Thank you for your positive assessment of the paper. Are there any concerns or confusions we  can help address? Please let us know, we are eager to help improve our paper in any way  possible.

---

### Official Review · Reviewer_F7yY · 2021-07-17

**Rating:** 7
**Confidence:** 5

**Summary:**

This paper presents a framework for teaching agents using language advice. The framework consists of three stages: (i) grounding language advice to action, (ii) learning an unconditioned policy that mimics the advice-conditioned policy (iii) evaluating the unconditioned policy. Experiments are conducted in maze navigation environments. Results demonstrate the effectiveness of the framework compared with imitation learning and reinforcement learning.

**Limitations And Societal Impact:**

The authors admit the limitations of the language and environments used in their experiments. The language is scripted and the environment is relatively simple (compared to the Matterport3D environment in [4]). I would like to point out several potential challenges when scaling this framework to more complex environments:
- Advice arises in contexts. In reality, we may not be able to ask humans to just "give me an advice" without grounding the question in the context of a task. Hence, it may make more sense to merge the grounding phase and the improvement phase as in [4], where humans give advice and also demonstrate the meaning of the advice at the same time so that the humans have contexts to give advice.
- Giving advice too frequently could be tedious for humans. To reduce human effort, for example, [4] learns a policy that allows the agent to actively decide when it is necessary to ask humans for advice.
- In complex environments, reinforcement learning could be too inefficient for language grounding. In that case, imitation learning could be used instead (as in [3, 4, 6]).




**Main Review:**

**Originality**: this work combines several pre-existing ideas, namely grounded language learning and policy distillation, in a novel way to enable teaching using language advice. It shares a lot of similarities with [6] (in hierarchical learning) and [3,4] (in learning to interpret advice) but the specific frameworks are not exactly the same. I would like to request comparison with these works in the paper and in the rebuttal. I also suggest reading [2] for a discussion on the drawbacks of reinforcement learning and imitation learning compared to a language-based learning framework.

Missing related work:

[1] Learning Rewards from Linguistic Feedback. https://arxiv.org/pdf/2009.14715.pdf

[2] Interactive Learning from Activity Description. https://arxiv.org/pdf/2102.07024.pdf

[3] Vision-based navigation with language-based assistance via imitation learning with indirect intervention. https://arxiv.org/pdf/1812.04155.pdf

[4] Help, Anna! Visual navigation with natural multimodal assistance via retrospective curiosity-encouraging imitation learning. https://arxiv.org/pdf/1909.01871.pdf

[5] Vision-and-Dialog Navigation. https://arxiv.org/pdf/1907.04957.pdf

[6] Hierarchical Imitation and Reinforcement Learning. https://arxiv.org/pdf/1803.00590.pdf

In fact, I strongly suggest the authors add more citations from the related work sections of these papers.

**Quality**:

The method is technically sound as it combines well-established methods. My concerns are mostly about evaluation:
- Using advice unit to measure communication efficiency oversimplifies comparison. A piece of language advice contains a lot of more information than a scalar reward. A scalar reward is represented by a 64-bit floating-point number in most machines; a 5-word advice in which each word is drawn from a 10^5-word vocabulary can be represented with log(10^5) * 5 ~ 80 bits; an action in a space consisting of 4 actions can be represented by only 2 bits. The bottom line is that communication efficiency of an advice should be based on the amount of information conveyed in the advice. A more sophisticated metric is to add the cognitive effort of humans when eliciting the advice but that may be too difficult to measure. Nevertheless, the advice unit metric in this work is overly naive and I do not advocate using it as a standard metric for later work to follow.
- The results for imitation learning are absurdly low. Is it because the training set is too small? I request the authors to describe the dataset used for training with this baseline. I strongly suggest using "behavior cloning" instead of "imitation learning" because the latter is a much broader term. In fact, when the advice is an Action Advice, the approach is equivalent to DAgger [7], which is a form of imitation learning. I encourage the authors to clarify this point in the paper.

[7] A Reduction of Imitation Learning and Structured Prediction to No-Regret Online Learning. https://www.ri.cmu.edu/pub_files/2011/4/Ross-AISTATS11-NoRegret.pdf

**Clarity**: The paper is well-written and is easy to follow. However, I would like to suggest the authors merge the "bootstrapping with multi-level advice" with section 4.3 because they implement the same idea. In fact, this framework would be more general if the authors adapted a goal-conditioned RL framework (e.g. [8]). Then, this framework can be described as low-level policies being composed to form higher-level policies until the highest-level policies are learned.

[8] Hindsight Experience Replay. https://arxiv.org/pdf/1707.01495.pdf

**Significance**: The paper studies an important direction: learning from language interaction, which enables non-expert humans to teach ML-based agents using their natural mode of communication. The presented framework is relatively general and can be potentially applied to a diverse range of tasks, especially that it captures the hierarchical structure of linguistically described tasks.



**Time Spent Reviewing:**

4 hours

---

> ### Author Response · Authors · 2021-08-10
> **Author's Response**
>
> Thank you for your suggestions and comments on our work. Please find detailed responses below:
>
> **“Missing related work” >** Thank you for pointing us to these works! We have added a discussion of each of these to the paper draft as follows: works such as [3] and [4] operate in a slightly different scenario where the final policy learned is advice-conditional rather than advice-free, like in our setting. These papers introduce the useful idea of actively querying for advice, which we could be incorporated into our framework.  [5] also operates in a different setting without in-the-loop advice. [1], like our work, uses human advice to supervise an agent, but their method is restricted to language advice whereas ours handles arbitrary prescriptive advice. We will address [2], [6], [7], and [8] in more depth in the additional experiments we have run related to them (please see the shared response). We have also added in further citations from the related work sections of these papers as suggested.
>
> We have also conducted new experiments comparing to  approaches using hierarchical RL and language  guided RL. Please refer to the shared response for a more detailed discussion and to experiment details here: https://tinyurl.com/teachable-rebuttal. In short, the experiments we conducted include:
>
> **HRL comparison:** in this extension to our method, we use the advice to supervise a high-level policy in a HRL framework. Experiments are in progress.
>
> **Language-guided RL comparison:** We compare against the efficiency of providing hindsight goal relabeling supervision.  Like in [2, 8], we collect trajectories and then relabel goals with a goal the agent actually achieved. We find that this method only allows the agent to learn in simple environments and is less efficient than using advice supervision.
>
> **“Using advice unit to measure communication efficiency oversimplifies comparison” >** Thank you for pointing this out, this is a very fair point. The goal of this metric is to capture the intuition that some forms of supervision are more time-intensive for a human to provide than others. In the scripted teacher experiments in the paper, we believe that the “advice unit” metric, while imperfect, still better captures the differences in efficiency between advice forms than sample efficiency. In our real human experiments (see https://tinyurl.com/exp1-human), we measure the amount of time it takes a human to provide supervision, which is a more reliable method, and we also compare sample efficiency in the supplementary materials section D.
>
> **"The results for imitation learning are absurdly low." >:**
> The behavioral cloning (BC) results are shown in Figures 7 and 20 (in the supplementary). (Note that the BC curve is missing a legend label in Fig 20 but is the gray triangles.), We will note in the paper that these are results for behavior cloning more specifically.
>
> In Fig 7, the BC plots for the PointMaze env are the second highest grey line (also shown more clearly here https://tinyurl.com/bc-curves). We note that the same exact code was used for both advice distillation and imitation learning, the only difference being the training data for imitation. We believe that it’s harder for imitation learning to perform well and generalize in this case because the demonstrations are generated from a hard-coded oracle policy, without significant variation or noise, which makes imitation learning less effective than if collected from a noisy RL policy. We don’t show BC results for AntMaze (where we have no oracle) or BabyAI (because providing Action Advice to a well-grounded policy (yellow curve in Fig 7c) is so similar to providing a demo that the results are nearly identical).
>
> **“When the advice is an Action Advice, the approach is equivalent to DAgger”>**
> These approaches aren’t quite equivalent. In DAgger, the agent completes a full trajectory before receiving optimal action labels. In our framework, on the other hand, the human provides Action Advice in the loop which the surrogate policy is able to directly condition on. To understand a variant of our approach which is equivalent to DAgger when providing Action Advice, see here: https://tinyurl.com/exp3-dagger
>
> **Limitations:** Thank you for pointing out these challenges. While we believe this is out of the scope of this work itself, we definitely agree these are all challenges that arise as we scale, and we will add a discussion of this.
>
> [1] Learning Rewards from Linguistic Feedback. https://arxiv.org/pdf/2009.14715.pdf
>
> [2] Interactive Learning from Activity Description. https://arxiv.org/pdf/2102.07024.pdf
>
> [3] Vision-based navigation with language-based assistance via imitation learning with indirect intervention. https://arxiv.org/pdf/1812.04155.pdf
>
> [4] Help, Anna! Visual navigation with natural multimodal assistance via retrospective curiosity-encouraging imitation learning. https://arxiv.org/pdf/1909.01871.pdf
>
> [5] Vision-and-Dialog Navigation. https://arxiv.org/pdf/1907.04957.pdf
>
> [6] Hierarchical Imitation and Reinforcement Learning. https://arxiv.org/pdf/1803.00590.pdf
>
> [7] A Reduction of Imitation Learning and Structured Prediction to No-Regret Online Learning. https://www.ri.cmu.edu/pub_files/2011/4/Ross-AISTATS11-NoRegret.pdf
>
> [8] Hindsight Experience Replay. https://arxiv.org/pdf/1707.01495.pdf

---

> > ### Comment · Reviewer_F7yY · 2021-08-27
> > **Happy with the response!**
> >
> > Thank you very much for the thorough response. My concerns are mostly addressed. I agree with the author about the difference between providing the action advice and DAgger. I also greatly appreciate the experiments with real humans, although I will need to double-check the AC that you are allowed to put out additional results in the rebuttal. Also, these results are great but need to be read with caution because of the simplicity of the tasks.
> >
> > My only remaining concern is about the advice unit metric. I will raise the score if the authors promise to highlight the limitations of this metric, discuss the challenges of constructing a fair automatic metric for measuring teaching effort, and emphasize the importance of experimenting with real humans.

---

> > > ### Author Response · Authors · 2021-08-28
> > > **Agree with your point on advice metrics**
> > >
> > > We’re glad we were able to address your concerns! We understand your point about the simplicity of the tasks, and we are in the process of addressing this at least partially by running the additional experiments we showed in the rebuttal on the full set of environments we have been using (PointMaze, Ant, BabyAI, multiple mazes for each). These results will be included in the final version of the paper.
> > >
> > > We also agree that the “advice unit” metric which we used in our simulated experiments is limited; while it does capture the fact that sparser forms of advice are often easier to provide, it doesn’t capture the fact that some advice units may be more natural or faster for a human to provide. We’ll add a discussion to the paper of this and highlight the value of directly measuring how much time it takes for a real human to provide advice through human experiments.

---

> > > ### Author Response · Authors · 2021-09-10
> > > **Have we addressed all concerns?**
> > >
> > > We hope we have addressed your concerns sufficiently with our response and we hope you will raise the score accordingly. And please let us know if there are other things we can do to address your concerns!

---

> > > > ### Comment · Reviewer_F7yY · 2021-09-10
> > > > **Raised the score!**
> > > >
> > > > I Raised the score to 7!

---

> > > > ### Comment · Reviewer_TZyF · 2021-09-10
> > > > **Maintaining original Score**
> > > >
> > > > I am maintaining my original score of 7.

---

### Author Response · Authors · 2021-08-10
**Author's Response + Additional Experiments**

We thank the reviewers for their thoughtful comments and feedback! We appreciate the positive reception of our novel problem setting and the importance of this training paradigm. We have run a number of new experiments to address reviewer comments. Replies to each reviewer concern can be found as responses to each individual review.

More details on all experiments can be found here: https://tinyurl.com/teachable-rebuttal

**Experiment 1: Real-Human experiments.** This experiment addresses concerns by Reviewers F7yY, TZyF, and df4E. We ran similar experiments to Section 5.4, Fig 7, but replaced the scripted teacher with a real human providing advice through a click interface on a screen. In the BabyAI environment, we found that distilling from advice was 67% more effective than providing a demonstration, accomplishing nearly 100% success on a task which required a new skill (unlocking a door). In the Ant environment, we found that with < 30 mins of human coaching time, the agent can reach 60-80% success, whereas an agent trained with dense-reward RL performs worse even after 4M more samples. Performance could be further improved if the human provided advice for longer. Details can be found at https://tinyurl.com/exp1-human.

**Experiment 2: Adding noise.** We added advice noise during the improvement phase as suggested by Reviewer df4E, and found that performance degrades only slightly, even with 30% of the advice provided containing noise. Details can be found at https://tinyurl.com/exp2-noise. Additionally the human experiments linked indicate a similar pattern with noisy human advice.

**Experiment 3: DAgger relabeling.** To address comparisons requested by Reviewer F7yY, we adapted a modified dataset aggregation or DAgger procedure to our problem setting. In particular, we experimented with having the teacher label completed trajectories with advice after the fact during the improvement phase, and then using the advice conditioned policy q_phi to relabel the trajectory with optimal actions. This relabeled data is then used to train the advice independent policy pi_theta, as described in [1] and [2]. A full description can be found at https://tinyurl.com/exp3-dagger.This DAgger-like scheme for soliciting advice performs comparably to receiving real-time advice on the PointMaze and AntMaze environments, and it removes the need for a human to be constantly present in the loop. This can be seen as an extension to the technique described in the current manuscript which allows for agents to be provided advice asynchronously with post-hoc human supervision.

**Experiment 4: Comparison to methods for hierarchical RL:** This ongoing experiment addresses points made by Reviewers F7yY and df4E. We explore a variation of our method where advice is used to supervise the high level of a hierarchical policy. Details can be found here: https://tinyurl.com/exp4-hrl. We find that in the AntMaze environment, by using a hierarchical model with our method, we can improve performance at convergence by 30% with fewer samples than needed to train purely a flat policy.

**Experiment 5: Comparison to methods for hindsight relabeling:** This experiment performs comparisons suggested by Reviewer F7yY. Based on the technique in [3], rather than initially training with RL, we roll out trajectories and hindsight relabel the agent’s goal with the goal that was actually achieved. This lets us then treat this data as optimal and train with supervised learning (as also described in [4]). We find that hindsight relabeling is only an effective supervision signal in simple environments (achieving close to 100% success on simple environments but < 40% on more challenging environments), and in all cases is less sample-efficient than advice distillation. Results can be found here: https://tinyurl.com/exp5-hindsight

**Behavioral Cloning and Sample Efficiency comparisons:** These plots show comparisons requested by Reviewers F7yY and TZyF. We present additional plots highlighting comparisons to a BC baseline (in addition to those presented in Figures 7 and 20), and additional sample efficiency comparisons (in addition to those shown in Appendix D). We see that behavioral cloning typically outperforms RL training but underperforms distilling from advice. Plots and discussion of these results can be shown here: https://tinyurl.com/bc-curves.

Citations:

[1] A Reduction of Imitation Learning and Structured Prediction to No-Regret Online Learning. https://www.ri.cmu.edu/pub_files/2011/4/Ross-AISTATS11-NoRegret.pdf

[2] Hierarchical Imitation and Reinforcement Learning. https://arxiv.org/pdf/1803.00590.pdf

[3] Interactive Learning from Activity Description. https://arxiv.org/pdf/2102.07024.pdf

[4] Learning to reach goals via Iteratived Supervised Learning https://arxiv.org/abs/1912.06088

---

> ### Author Response · Authors · 2021-08-16
> **Update**
>
> Two notes to reviewers:
>
> (1) The first plot in Experiment 3 has been updated to correct a plotting error. The update shows improved DAgger results and doesn't change the overall experiment conclusion that DAgger relabeling is a promising extension to our method.
>
> (2) Please let us know if there are any questions or concerns you have remaining about our paper. We're eager to clarify and address your concerns!

---

### Author Response · Authors · 2021-08-26
**Do reviewers have additional concerns?**

Please let us know if we've addressed your concerns. We'd be happy to explain or discuss any questions you have remaining.

---

### Decision · Program_Chairs · 2021-09-27

**Decision:**

Accept (Poster)

**Comment:**

The reviewers were very impressed by the amount of additional material provided during the author response phase. The extensive replies addressed all major concerns of the reviewers. Incorporating all that in the paper will be a challenge though.